# Dynamic genome-based metabolic modeling of the predominant cellulolytic rumen bacterium *Fibrobacter succinogenes* S85

Ibrahim Fakih,[1,2] Jeanne Got,[3] Carlos Eduardo Robles-Rodriguez,[4] Anne Siegel,[3] Evelyne Forano,[1] Rafael Muñoz-Tamayo[2]

**ABSTRACT**  *Fibrobacter succinogenes* is a cellulolytic bacterium that plays an essential role in the degradation of plant fibers in the rumen ecosystem. It converts cellulose polymers into intracellular glycogen and the fermentation metabolites succinate, acetate, and formate. We developed dynamic models of *F. succinogenes* S85 metabolism on glucose, cellobiose, and cellulose on the basis of a network reconstruction done with the automatic reconstruction of metabolic model workspace. The reconstruction was based on genome annotation, five template-based orthology methods, gap filling, and manual curation. The metabolic network of *F. succinogenes* S85 comprises 1,565 reactions with 77% linked to 1,317 genes, 1,586 unique metabolites, and 931 pathways. The network was reduced using the NetRed algorithm and analyzed for the computation of elementary flux modes. A yield analysis was further performed to select a minimal set of macroscopic reactions for each substrate. The accuracy of the models was acceptable in simulating *F. succinogenes* carbohydrate metabolism with an average coefficient of variation of the root mean squared error of 19%. The resulting models are useful resources for investigating the metabolic capabilities of *F. succinogenes* S85, including the dynamics of metabolite production. Such an approach is a key step toward the integration of omics microbial information into predictive models of rumen metabolism.

**IMPORTANCE**  *F. succinogenes* S85 is a cellulose-degrading and succinate-producing bacterium. Such functions are central for the rumen ecosystem and are of special interest for several industrial applications. This work illustrates how information of the genome of *F. succinogenes* can be translated to develop predictive dynamic models of rumen fermentation processes. We expect this approach can be applied to other rumen microbes for producing a model of rumen microbiome that can be used for studying microbial manipulation strategies aimed at enhancing feed utilization and mitigating enteric emissions.

**KEYWORDS**  dynamic model, elementary flux mode analysis, genome-scale metabolic model, fiber degradation, network reconstruction, rumen fermentation

The rumen microbiota plays an essential role in ruminant nutrition by breaking down and fermenting plant-based feed, transforming it into a source of energy and protein for the host. The rumen microbiota is composed of a very diverse community of prokaryotes (bacteria and archaea) and eukaryotes (protozoa and fungi) which concur to the degradation and fermentation of the feed components, and particularly complex fibrous substrates that cannot be digested by the host. Rumen bacteria, fungi, and protozoa participate in the degradation of the plant cell wall lignocellulose (1), producing a large array of enzymes and various enzymatic systems to deconstruct the intricate chemical structure of plant biomass (2). Among them, cellulose degraders

Address correspondence to Rafael Muñoz-Tamayo, rafael.munoz-tamayo@inrae.fr.

The authors declare no conflict of interest.

See the funding table on p. 20.

have been particularly studied for decades, because cellulose is the most degradation-resistant polysaccharide in plants, and it represents an abundant renewable resource on earth (3). Within cellulolytic bacteria, *Fibrobacter succinogenes* has been particularly studied (2). *F. succinogenes* is found in large numbers in ruminants fed high-fiber diets (4) and is present in domestic and wild ruminant species from many geographical regions worldwide (5). It has been quantified at higher levels in bovines compared to deer, sheep, or camelids, suggesting that it may play an essential role in plant fiber degradation in cattle. *F. succinogenes* belongs to the *Fibrobacteres* phylum which also comprises the species *Fibrobacter intestinalis*, mainly isolated from the feces of ruminant and non-ruminant animals (6).

The strain *F. succinogenes* S85 has been isolated from a bovine rumen a long time ago (7, 8) and is the most studied strain of the species. For efficient plant cell wall degradation, *F. succinogenes* adheres closely to the substrate and produces specific cellulose-binding proteins and possibly also pili to mediate its adhesion (9–11). *F. succinogenes* is considered as particularly efficient in the hydrolysis of crystalline cellulose, and it degrades at the same rate amorphous and crystalline regions of wheat straw cellulose (12). Cellulose is degraded into cellodextrins, cellobiose, and glucose, and *F. succinogenes* was shown to be a very effective competitor for cellodextrin utilization (13). The bacterium is also able to synthesize and efflux oligosaccharides that may be used by other rumen bacteria through cross-feeding (12, 14). Given all these properties, it may be interesting to promote *F. succinogenes* populations in the rumen of cattle to improve the degradation of recalcitrant substrates and their utilization by the rumen microbiota.

The analysis of the *F. succinogenes* S85 genome showed that it consists of approximately 3.84 Mbp with a GC content of 48% and that it contains a high number of genes (134) encoding carbohydrate-active enzymes (CAZymes) (10). The genome analysis also confirmed that despite its ability to degrade xylans (15), *F. succinogenes* cannot use xylose because it lacks the sugar transporter and phosphorylation system (16). This species is thus a cellulose specialist using cellulose and its degradation products as its sole energy source. Glucose and cellobiose are fermented mainly through the Embden-Meyerhof-Parnas (EMP) pathway into succinate as major final product, followed by acetate, formate and $CO_2$. *F. succinogenes* is able to store intracellular glycogen which can represent up to 70% of the dry weight of the bacterium (17). This storage could allow bacteria to remain in the rumen in the absence of metabolizable substrates (18), but the intracellular glycogen is simultaneously stored and degraded, suggesting a futile cycling (19). *F. succinogenes* uses ammonia as the sole source of nitrogen, and several steps in the ammonia assimilation pathway have been identified (20). In addition to its interest for ruminant nutrition, *F. succinogenes* has also received much attention from the biotechnology sector (21). Firstly, because this species produces an original cellulolytic system, whose organization is still not well understood, and includes membrane vesicles as vehicles of CAZymes (22). Deciphering this system could help in the design of novel Consolidated Bioprocessing (CBP) for the production of cost-effective and sustainable lignocellulosic biofuels (9). Secondly, the capacity of *F. succinogenes* to transform lignocellulosic material into succinate may also be of interest because succinic acid could be used as a platform molecule (23, ,24). However, increasing bacterial product yield or maximizing the production of powerful lignocellulose degradation enzymes is dependent on detailed knowledge of metabolic pathways for microbial engineering processes (25). An efficient way of deciphering a bacterium metabolic network and identifying possible bottlenecks in the production of metabolites is *via* genome-scale metabolic models (GEM). A GEM is a mathematical representation of a metabolic network that allows the study of genotype-phenotype relationships (26) and facilitates the prediction of multiscale phenotypes (27). For a genome-sequenced microorganism, a GEM is defined by a stoichiometry matrix that links metabolites to the collection of reactions that occur in the organisms according to evidences about genes catalyzing the reactions. The resulting metabolic network can be further analyzed using methods such as flux balance analysis (FBA) (28–30). However, the FBA approach

does not allow to predict the dynamics of metabolites concentrations. In parallel, kinetic modeling approaches allow to represent the dynamics of metabolites of interest by deriving mass balance equations (31). Kinetic models are built following a macroscopic representation of the metabolism with a reduced set of macroscopic reactions which are often selected from documented literature. These models are called unstructured models where the cell biomass is described by a single state variable in addition to the concentration of extracellular metabolites [see reference (32) for a review on kinetic models]. Kinetic unstructured models rarely integrated microbial genomic information, with the exception of certain approaches that exploit the information of the stoichiometry matrix to derive macroscopic reactions (33). The objective of this work was to develop dynamic metabolic models (DMM) to represent the metabolism of glucose, cellobiose and cellulose by of *F. succinogenes*. These DMM integrate microbial genomic knowledge from the reconstruction of a GEM of *F. succinogenes*.

## MATERIALS AND METHODS

### Culture conditions and sample preparation

*F. succinogenes* strain S85 (ATCC 19169) was grown in triplicates in a chemically defined medium (19) with 3 g/L glucose, cellobiose, or filter paper cellulose. The culture medium used the mineral medium of Bryant and Burkey (34) as a base, with the following additions: volatile fatty acids (mM): acetic acid 28.8, propionic acid 7.93, *n*-butyric acid 4.27, isovaleric acid 1.27, DL- α-methyl butyric acid 0.9, isobutyric acid 1.02, *n*-valeric acid 1.3, resazurin 1 mg/L, *p*-aminobenzoic acid 0.1 mg/L, biotin 0.05 mg/L, hemin 1 mg/L, and cysteine/HCl 0.5 g/L.

The cultures were grown at 39°C using Hungate tubes and the Hungate anaerobic cultivation technique, under a 100% $CO_2$ atmosphere (35). The bacterial growth on cellobiose or glucose was monitored by measuring the absorbance at 600 nm. The quantification of succinate was used to monitor growth on cellulose cultures (12, 36). During growth, supernatants were collected by centrifugation (10,000 rpm for 5 min at 4°C), at six time points of the bacterial growth (0, 5, 9, 13, 17, 20, and 24 hours), and then stored at −20°C for further analysis.

### Quantification of substrate consumption and metabolite production

The triplicate culture supernatants at each time point were used for substrate and end-product assays. Concentrations of succinate, acetate, formate, ammonia, and glucose were measured in culture supernatants by enzymatic methods, using Megazyme kits (K-SUCC 06/18, K-ACET 04/18, K-FORM 10/17, K-AMIAR 04/18, and K-GLUHK-220A, respectively) according to the manufacturer's recommendations. Cellobiose consumption was estimated by quantification of the remaining reducing sugars in the culture medium using Miller's method (37).

### Metabolic network reconstruction

The metabolic reconstruction of *F. succinogenes* S85 was performed using the freely available workspace AuReMe (*au*tomatic *re*construction of *me*tabolic models) (38). AuReMe embeds existing tools as well as *ad hoc* packages to reconstruct and handle GEMs. It uses (i) the outputs of the Pathway Tools software (39, 40) to perform annotation-based reconstruction, (ii) the OrthoFinder method (41, 42) to perform orthology-based reconstructions, and (iii) the Meneco tool (43) to perform a gap-filling procedure. AuReMe relies on the PADMet library (*P*ython library for h*a*ndling meta*d*ata of *met*abolism [38]) to ensure the reproducibility of the workflow used to create a GEM, to guaranty the interoperability between the different methods used, to curate GEMs, and to export the GEM under several formats (SBML, matrix, wiki, RDF-like format). AuReMe also uses CobraPy (44), a Python package to analyze FBA, and flux variability analysis (FVA)

(analysis of essential and blocked reactions). In Fig. 1, we summarize the steps of the reconstruction of *F. succinogenes* S85 metabolic network.

### Step 1 (collecting genomes and reference reaction data set)

The two complete annotated genomes of *F. succinogenes* S85 were downloaded, respectively, from https://www.ncbi.nlm.nih.gov/nuccore/CP001792.1 and https://www.ncbi.nlm.nih.gov/nuccore/CP002158.1. Each *F. succinogenes* S85 genome contains 3.84 Mbp with, respectively, 3,160 and 3,174 genes identified.

The biomass reaction of *Escherichia coli* K-12 MG1655 (45) was adapted to *F. succinogenes* and used to build the metabolic network (Table S1 in supplementary material A; the supplemental material is available at https://doi.org/10.5281/zenodo.7228115).

The list of seeds (essential constituents of the culture medium to guarantee growth) was prepared based on the minimal medium composition needed for *F. succinogenes* growth (Table S2). The final products' (targets') list was prepared according to our knowledge on the metabolism of this bacterium metabolism and network reconstruction needs (Table S2).

### Step 2 (generating draft models)

A first GEM was reconstructed according to the genome annotations via Pathway Tools using both complete genomes: NC_013410.1 and NC_017448. In parallel, orthology-based reconstruction GEMs were obtained using the GEMs of the following gut microbes: *Bacteroides thetaiotaomicron* iAH991 (46), *E. coli* K-12 MG1655 (45), *Faecalibacterium prausnitzii* A165 (29), *Bifidobacterium adolescentis* L2-32 (28), and *Lactobacillus plantarum* WCFS1 (47), and mapped to MetaCyc (48), thanks to the MetaNetX database (49). Finally, all GEMs obtained from the annotation and the orthology reconstruction steps were combined into a draft GEM with the PADMet library.

### Step 3 (gap filling)

To analyze and curate the GEM of *F. succinogenes* S85, we applied a gap-filling procedure. Here, a GEM is considered as a graph in which metabolites are nodes and reactions are the links between the nodes. In these analyses, stoichiometry is not considered. This procedure allows adding reactions to guarantee the production of specific metabolites according to a graph-expansion criterion. We used Meneco (*me*tabolic *ne*twork *co*mpletion [43]) and the MeneTools package (*me*tabolic *ne*twork *to*pological to*ols* [38, 50]) for this gap-filling step.

### Step 4 (manual curation and gene-reaction association)

The draft network was manually curated to find potential errors and filling gaps based on the phenotype and experimental data reported in the literature. FBA was used to reconstruct and validate models for maximizing the biomass reaction flux.

Some manually and gap-filled added reactions had no gene associated. All the gene sequences from other bacteria associated with these reactions were identified in the National Center for Biotechnology Information (NCBI) using the reaction Enzyme Commission (EC) number. The corresponding protein sequences were aligned using BLAST (*b*asic *l*ocal *a*lignment *s*earch *t*ool [51]) to the *F. succinogenes* S85 translated genomes. The identified proteins with identity >76% with coverage throughout the sequence were associated with their corresponding reactions in the GEM. We set a high identity percentage to avoid biological inconsistencies by linking a gene to a reaction without experimental validation. Reactions with no gene associated or with gene coding protein of lower similarity were retained in the model only when present in the *E. coli* K-12 MG1655 model (45).

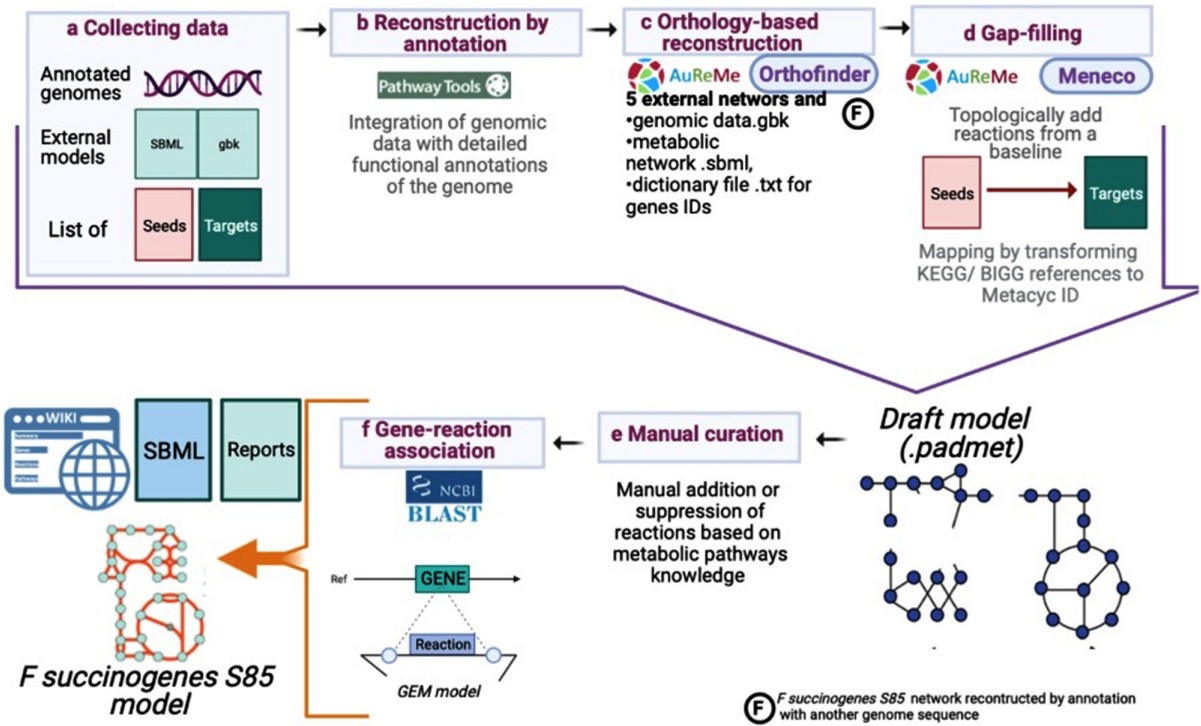

**FIG 1** *Fibrobacter succinogenes* S85 GEM reconstruction pipeline (created with BioRender).

## Construction of a dynamic metabolic model

### *Exploiting EFMs to derive a macroscopic dynamic metabolic model*

The dynamics of metabolism can be described by the following generic differential equation resulting from applying a mass balance (equation 1)

$$\frac{d\mathbf{x}}{dt} = \mathbf{S}\,\mathbf{r}(\,\cdot\,) \tag{1}$$

where $\mathbf{x}$ is the vector containing the concentrations of metabolites, which can be either intracellular ($\mathbf{x}_i$) or extracellular ($\mathbf{x}_e$). The vector $\mathbf{r}(\,\cdot\,)$ represents the reaction rates, which are the function of the concentrations $\mathbf{x}$ and a parameter vector. The stoichiometric matrix $\mathbf{S}$ contains the stoichiometric matrices for intracellular ($\mathbf{S}_i$) and extracellular ($\mathbf{S}_e$) metabolites. Under the assumption that intracellular metabolism operates at a steady state, it follows that

$$\frac{d\mathbf{x}_i}{dt} = \mathbf{S}_i\,\mathbf{r}(\,\cdot\,) = \mathbf{0} \tag{2}$$

The vectors of reaction rates that fulfill (equation 2) are non-negative vectors contained in the null space of the stoichiometric matrix $\mathbf{S}_i$. The space of admissible fluxes is a convex polyhedral cone. The generating vectors of the cone are called elementary flux modes (EFMs). Any steady-state flux distribution can be expressed as a non-negative linear combination of the EFMs. Biochemically, EFMs are independent minimal pathways of the metabolic network that can operate at a steady state. Each EFM can be converted into a macroscopic reaction that connects extracellular substrates and products (33, 52). The identification of macroscopic reactions is the core of kinetic modeling. Once we find a set of macroscopic reactions to represent the metabolism of our microorganism of interest, we can derive the structure of a DMM.

For the $j$th EFM $\mathbf{e}_j$, the macroscopic reaction $j$ is obtained by the product $\mathbf{S}_e\,\mathbf{e}_j$. To calculate the EFMs of the network of *F. succinogenes*, we used the efmtool algorithm (53) of the MATLAB package CellNetAnalyzer (54), which is freely available at http://www2.mpi-magdeburg.mpg.de/projects/cna/cna.html. Then, we proceeded to select a reduced number of EFMs using the yield analysis method proposed by Song and Ramkrishna (55). The selected EFMs were expressed as macroscopic reactions to define further the kinetics in the dynamic metabolic modeling.

### Reduction of the GEM

The calculation of EFMs is restricted to medium-scale GEM (less than 350 reactions) (56). Hence, a complete EFM analysis of the network of *F. succinogenes* S85 is intractable. A reduction of the network is thus here proposed. Several methods for the reduction of GEM have been reported in the literature (57), which are mainly based on a fully functional core metabolic network that preserves a set of important moieties and capabilities from the full network. However, the selection of a subset of reactions might produce loss of information regarding parallel pathways that can be used to attain the same metabolic goal. In this work, we have selected another method called NetRed (58), which analyses flux vectors generated from the complete network (FBA) and computes a reduced network that holds the same flux distribution.

Differently from other methods, NetRed performs the reduction of the stoichiometric matrix of the full network through matrix algebra based on given flux vectors and a list of protected metabolites (57). This method provides a reduced network and its correspondent flux vector, which is consistent with the flux vector of the full stoichiometric matrix. NetRed is implemented in the MATLAB COBRA toolbox (59). This is advantageous since the results from FBA, also implemented in the COBRA toolbox, can be directly used for the reduction. As NetRed can manage several flux vectors, the reduced network can represent various flux distributions and provide a unique biomass reaction that respects the given flux vectors. Additionally, the use of the matrix approach allows mapping the reactions in the reduced network into their corresponding ones in the full network.

The reduction of the GEM followed several steps: (i) calculation of fluxes by FBA, (ii) carbon balancing, (iii) compacted lumped biomass reaction, (iv) re-calculation of fluxes by FBA, and (v) network reduction.

To keep the flexibility of the full network, the flux distribution of the GEM was calculated by FBA considering different objective functions that maximize the production of specific metabolites (i.e., biomass, succinate, and acetate). Furthermore, several input fluxes were considered for the three substrates: glucose, cellobiose, and cellulose. Details about the values are described in the section Network reduction.

The results were analyzed in terms of yields for which all the obtained fluxes were divided by the uptake flux of glucose, cellobiose, or cellulose. Yield analysis allowed to verify the carbon balance of the network as well.

Further reduction of the network was achieved by computing a compact lumped biomass reaction, which was based on the pathways identified as essential for biomass production. This approach is like the construction of a core metabolism. In our case, however, we have only used this core for biomass allowing for other pathways to contribute as well to the production of metabolites needed for biomass. Finally, the obtained fluxes from the network with compacted biomass formulation and correct carbon balance have been introduced to NetRed to compute the final reduced network which was used to compute EFMs.

### Parameter identification

The model parameters were estimated from the *in vitro* experimental data detailed in the sections Culture conditions and sample preparation and Quantification of substrate consumption and metabolite production, where *F. succinogenes* was grown on three different substrates (glucose, cellobiose, and cellulose). During culture on these substrates, growth (optical density [OD] 600 nm), production of metabolites (succinate,

acetate, and formate), ammonia and carbohydrate consumption were monitored at six time points. The dynamic concentration of the metabolites were used in the model calibration routine using the maximum likelihood approach implemented in the MATLAB toolbox IDEAS (60), which is freely available at http://genome.jouy.inra.fr/logiciels/IDEAS. The optimization uses the quasi-newton algorithm implemented in the MATLAB function *fminunc*. The dynamic model representing the fermentation of each substrate is defined by the kinetic rate function of substrate utilization through the macroscopic reactions given by the EFMs. For glucose and cellobiose, we modeled the macroscopic reactions as Monod functions (equation 3):

$$\mu_i = \mu_{\max,\,i} \cdot \frac{s_i}{K + s_i} \cdot B \qquad (3)$$

where $s_i$ and $B$ are the molar concentrations of the substrate and biomass, respectively, $\mu_i$ is the microbial growth rate for the reaction $i$, $\mu_{\max,\,i}$ is the maximal growth rate constant (per hour), and $K$ is the substrate affinity Monod constant (moles per liter). The estimation of the parameters of Monod kinetics is known to be hampered by practical identifiability problems. That is, the parameters cannot be estimated uniquely from noisy and limited data (61). This situation is reflected by a high correlation between the model parameters. When practical identifiability problems occur, we should be cautious in providing interpretations from the numerical values of the parameter estimates (62). The interested reader in parameter identifiability aspects is referred to Muñoz-Tamayo et al. (63). To avoid the practical identifiability problems mentioned above in our case study, we set $K$ to $9 \times 10^{-3}$ M as in the rumen model developed by Muñoz-Tamayo et al. (64). Since cellulose is a particulate substrate, we modeled the macroscopic reactions using the Contois function (equation 4) as proposed by Vavilin et al. (65).

$$\mu_i = \mu_{\max,\,i} \cdot \frac{s_i}{K_c \cdot B + s_i} \cdot B \qquad (4)$$

where $K_c$ is the half-saturation Contois constant. For each substrate, we selected initially the EFMs that correspond to the vertices of the polygon enclosing the yield spaces. A further reduction was implemented within the calibration procedure by adding a penalization coefficient in the cost function of the optimization to penalize a large number of EFMs. To account for the death of microbial cells, we included a first-order kinetic rate with a death rate constant $k_d$ set to $8.33 \times 10^{-4}$ as in Muñoz-Tamayo et al. (64). We also included a conversion factor α (M/OD) to transform the biomass concentration (moles per liter) into OD. This conversion is needed to compare the model output of biomass against the measured OD. It should be noted that for cellulose, OD was not measured. For this case, the initial condition of biomass was included as a parameter to be estimated. We evaluated the model accuracy using the coefficient of variation of the root mean squared error (CVRMSE).

## RESULTS

Following Open Science practices to promote accessibility and reproducibility (66), the metabolic network and mathematical models developed in this work are freely available at https://doi.org/10.5281/zenodo.7228115.

### Description of the network

#### *Large-scale genome reconstruction process*

The two published genomes of the *F. succinogenes* S85 strain were used to identify potential reactions that could be present in the GEM of the bacterium. Genome annotation performed by Pathway Tools detected 827 reactions (Fig. 2A) and 1,112 metabolites. Eight hundred seventeen reactions are common between the two available

*F. succinogenes* S85 annotated genomes. Four and six reactions were specific to NC_013410 and NC_017448 genomes, respectively (Table S3 in supplementary material A), illustrating that the two genome sequences are not complete.

### Orthology and gap filling

First, we downloaded the annotated genome sequences of the five external models from NCBI and their GEM SBML format with the reference file of ID reactions present in KEGG, BiGG, or MetaCyc. Reconstruction by orthology provided 174 reactions for the *F. succinogenes* S85 first model obtained from Pathway Tools (Fig. 2A). In addition, 203 reactions were brought by the combination of reconstructions by annotation and orthology. Finally, 61% of orthology-based reactions were added to the network according to cross-sources (Fig. 2B).

The definition of seeds and targets is an essential step in the reconstruction protocol. We have determined a list of 51 sources (constituents of the culture medium) and 85 targets (molecules known to be produced by the bacterium) for *F. succinogenes* S85 (Table S2 in supplementary material A). The gaps in the model were first filled automatically by mapping and transforming KEGG/BiGG identifiers to MetaCyc IDs using the MetaNetX package. Ninety-nine reactions were added by Meneco (43), and all of them came from the MetaCyc database (48). Twenty-four reactions were removed according to expert validation (see below), and finally 75 reactions were added by gap filling.

### FBA for unblocking biomass and manual curation

In our reconstruction process, the aim was to obtain a functional high-quality network, which produces biomass yield. For this purpose, we first focused on reaching topologically all the targets according to the qualitative network-expansion criteria. Forty-one reactions were expertly added to the gap-filled model network. These reactions belong to four distinct databases: MetaCyc (48), KEGG (67), BiGG (68), and RHEA (69).

Afterward, we manually unblocked all the pathways referring to FBA analysis, which were in direct and indirect relationships with each component of the biomass reaction. We therefore focused on curating the network and the pathways involved in carbon and nitrogen metabolism as well as some essential cofactor biosynthesis. In particular, we checked the different pathways of glycolysis, the glycogen cycle (see below), and the short-chain fatty acids biosynthesis. This required adding 490 manually curated reactions in the pathways of the biomass compounds.

### Manual completion

*F. succinogenes* S85 model reactions are linked to 1,317 genes, leading to 77% of reactions associated with a gene (Fig. 3). Four hundred thirty reactions were not linked to any gene before validation. Of them, 133 are exchange, transport or spontaneous reactions. For all the other reactions, we search for the presence of a gene possibly associated using BLAST (Data set S1 in supplementary material B). Finally, 68 unlinked gene reactions were added by gap filling, 54 reactions that seemed inappropriate to the metabolism of the bacterium were suppressed, and 22 reactions were linked manually to their corresponding gene.

### Qualitative analysis of the *Fibrobacter succinogenes* S85 metabolic network

#### FBA and essential reactions

The obtained network has 1,317 genes, has 931 pathways, and is composed of 1,565 reactions, from which 1,211 are associated with genes (Table 1). The final network contains 1,586 unique metabolites and is available online as a wiki page at: https://gem-aureme.genouest.org/fsucgem/index.php/Fsucgem.

We investigate our final metabolic network using FVA. All 85 target components are reached topologically, 38.5% of the reactions are active, of which 137 are essential

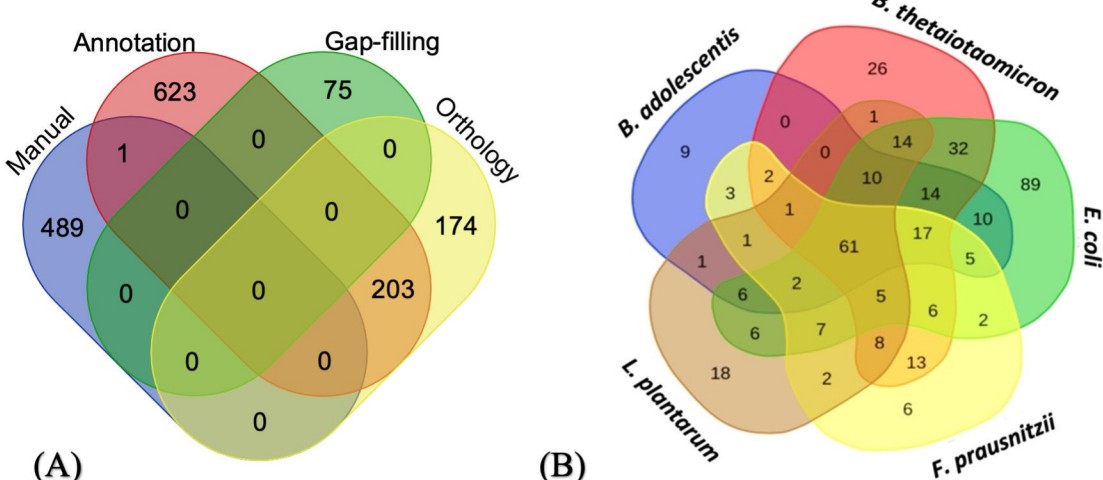

**FIG 2** Venn diagram representing (A) distribution of reactions across different steps of final model reconstruction and (B) distribution of reactions across various external metabolic models used for orthology-based model reconstruction.

reactions for biomass production. The simulated growth rate from *F. succinogenes* S85 metabolic model is 0.137/hour.

Three hundred eighteen out of 931 (34%) metabolic pathways are complete at more than 75% of the reactions present in the KEGG and MetaCyc databases. 76% of those active pathways have at least one reaction with flux according to the FBA analysis, and 150 pathways are 100% active in flow (Data set S2 in supplementary material B). All the essential reactions are present in the active pathways.

### Glycogen biosynthesis and degradation pathways

As an example to illustrate the use of the reconstructed network, we analyzed the glycogen biosynthesis and degradation pathways, because glycogen has been shown to be simultaneously synthesized and degraded in *F. succinogenes* during all growth phases (19). Intracellular glycogen accumulation is carried out by the consecutive action of ADP-glucose pyrophosphorylase (*glgC*, FISUC_RS14455 FSU_RS00645) (EC 2.7.7.27), glycogen synthase (*glgA* FSU_RS16140 FISUC_RS15965) (EC 2.4.1.21), and glycogen-branching enzyme (*glgB* FISUC_RS15575 FSU_RS01770) (EC 2.4.1.18) (Fig. 4). All these anabolism reactions and genes were identified by annotation, except the

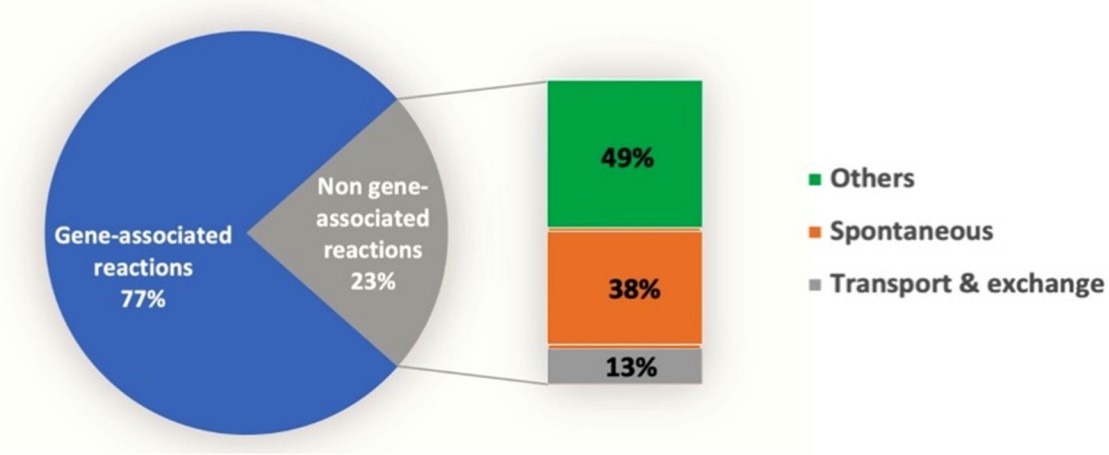

**FIG 3** Gene-reaction association.

**TABLE 1** *Fibrobacter succinogenes* S85 metabolic model information

| | *F. succinogenes* S85 model | | | |
|---|---|---|---|---|
| Reactions | 1,565 | | | |
| Unique metabolites | 1,586 | | | |
| Genes | 1,317 | | | |
| Active[a]/total pathways | 233/931 | | | |
| | No. of reactions[b] | | | |
| | **Annotation** | **Orthology** | **Gap filling** | **Manual curation** |
| | **Total: 827** | **Total: 377** | **Total: 75** | **Total: 490** |
| Exchange/transported | 1 | 15 | 0 | 133 |
| Spontaneous | 1 | 2 | 1 | 45 |
| Protein/amino acid biosynthesis | 101 | 74 | 16 | 25 |
| Glycolysis/fatty acid biosynthesis | 51 | 17 | 2 | 12 |

[a]Ratio (reaction found/total) >0.75 and ratio reaction with flux/reaction based on FBA analysis >0.5.
[b]Two hundred four reactions are identified from cross method sources.

phosphoglucomutase (*pgm* FSU_0773) that was linked to the reaction EC 5.4.2.2 by manual curation (Data set S1 in supplementary material B).

The known glycogen degradation pathway on MetaCyc 23.0 was detected with the presence of the reactions EC 2.4.1.1 (Maltotetraose glucosidase *malP* FSU_RS06195) and EC 2.4.1.25 (4-alpha-glucanotransferase *malQ* FISUC_RS04305; FSU_RS06200) identified by annotation (Fig. 4). Then, we completed this pathway by adding manually the two reactions EC 3.2.1.196 (limit dextrin α-1,6-glucohydrolase *glgX*) and EC 3.2.1.20 alpha-glucosidase (*malZ* FSU_RS06195). The maltotetraose formation reaction present in *E. coli* model was added to our model, and no gene was linked because of no significant BLAST similarity with the *F. succinogenes* genome (Data set S1 in supplementary material B).

## Network reduction

### Reduced-scale genome reconstruction model process

Network reduction was achieved by NetRed method, which is based on flux vectors computed by FBA from the full network. The reactions in the full network were defined as irreversible reactions by decoupling the reversible reaction into their forward and backward directions. The full network was then composed by 1,780 irreversible reactions. The measured metabolites obtained from batch cultures growing on glucose, cellobiose, and cellulose corresponded to extracellular acetate, formate, and succinate. From those results, it was observed that formate is produced in small amounts while succinate and acetate were the main products. Accordingly, two objective functions were defined to maximize concomitantly: (i) biomass and succinate and (ii) biomass and acetate. To enlarge the possible flux distribution, different input fluxes of glucose, cellobiose, and cellulose ranging between 0 and 1,000 mmol/$g_{biomass}$/hour with a step of 50 mmol/$g_{biomass}$/hour were considered. To facilitate the computation ease, we defined that cellobiose was composed of two molecules of glucose while cellulose was considered as four molecules of glucose, so the upper ranges were modified to 250 and 500 mmol/$g_{biomass}$/hour, respectively. The resultant fluxes were analyzed in terms of yields for which all fluxes were divided by the flux of the carbon uptake reactions.

Carbon balance was verified through yield analysis, where we have noticed that the glycogen synthesis and degradation pathways generated unbalanced carbon production. For the sake of carbon quantification, we assumed that glycogen and (1,4-α-D-glucan)$_n$ were composed of six and five molecules of glucose, respectively. Additionally, a new hypothetical reaction (equation 5) was added to consider the production of (1,4-α-D-glucan)$_n$ as

$$5 \text{ glc-1-P}_{[c]} = 5 \text{ P}_{i[c]} + 11\text{-4-alpha-D-glucan\_n}_{0[c]} \tag{5}$$

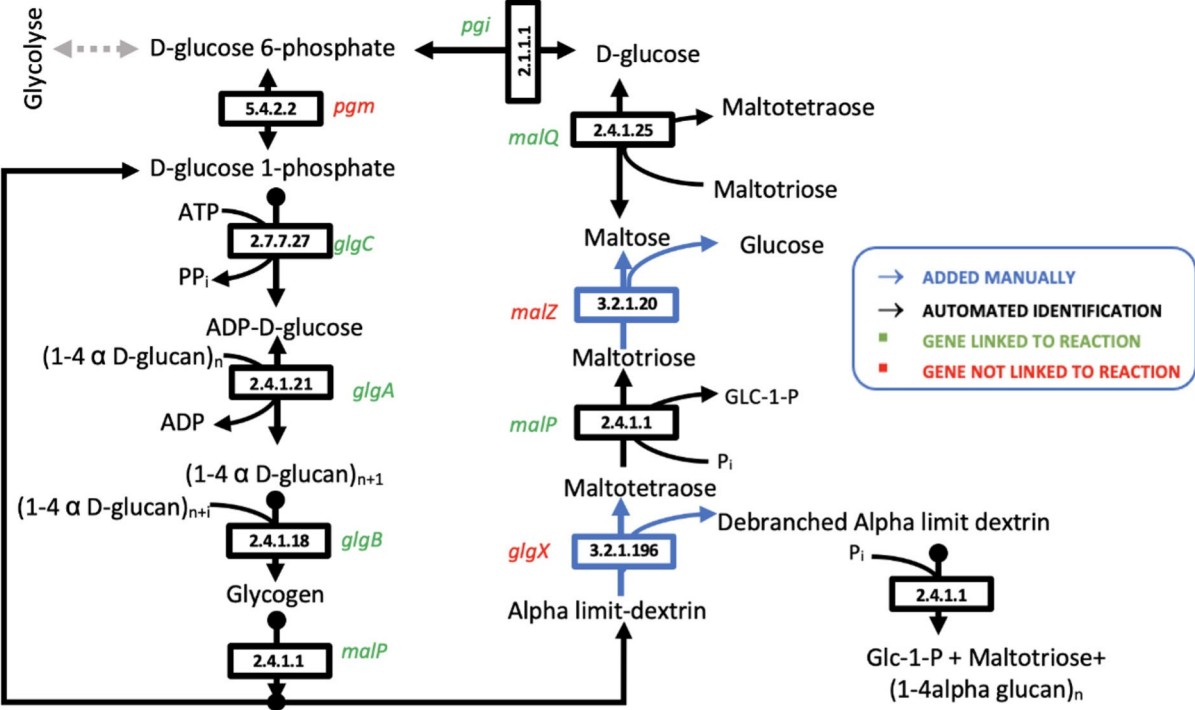

**FIG 4** Glycogen synthesis and degradation pathways identified and completed in *Fibrobacter succinogenes* S85 metabolic network.

This new reaction allowed to balance carbon and produce biomass with a mass of 25.394 g/mol$_C$. FBA was performed again to compute the flux vector and the reduction was done with NetRed. The protected metabolites corresponded to biomass, acetate, succinate, formate, glycogen, glucose, cellulose, cellobiose, protons, ammonium, and fructose-6P. The reduced network contained 146 reactions and 78 metabolites whose size was still large for the computation of EFM.

A further reduction of the network was achieved by constructing a lumped biomass reaction, which was based on the precursors of several metabolites such as amino acids. The precursors were identified by tracking back the pathways that produce the metabolite to be deleted from the 137 essential reactions identified for the full network. All the reactions involved in the pathways were added up to obtain a reaction to replace the metabolite to be deleted by its precursors (e.g., pyruvate, fructose-6P). This approach is similar to the computing of lumped biomass employed by Lugar et al. (58). Details about the construction of the lumped biomass could be found in Data set S3 in supplementary material B. Once the lumped biomass reaction was obtained, the coefficients were corrected to obtain a biomass of 26.401 g/mol$_C$ following the formula $C_{3.69}H_{6.76}O_{2.66}N_{0.25}S_{0.010}$ previously reported for the biomass composition of *F. succinogenes* S85 (70).

The new network comprising the lumped biomass reaction and the carbon balance was used to calculate flux vectors subject to several constraints. All the reactions used for the construction of the lumped biomass reaction that did not correspond to essential reaction were blocked to zero flux. Furthermore, we verified that cofactors such as FAD (Flavin adenine dinucleotide), P$_i$, PP$_i$, and ADP were not needed as sources, so their fluxes were also assumed to be zero. A reduction of the extracellular cofactors was made accepting small changes in yield analysis from FBA. Cofactors such as NADPH, NADP, NAD, and NADH were not needed as sources, whereas ADP and NADH were not needed as sinks.

Flux vectors for the three substrates and the protected metabolites mentioned before were used in NetRed to obtain a reduced network of 63 reactions with 36 intracellular metabolites and 16 extracellular metabolites. The biomass reaction of

**TABLE 2** Yield boundaries of the reduced network

|  | Yield | Biomass | Acetate | Succinate | Formate |
|---|---|---|---|---|---|
|  | Minimum | 0.0002 | 0.0000 | 0.0000 | 0.0000 |
| Glucose | Maximum | 0.1465 | 2.6231 | 1.5622 | 3.8813 |
|  | Minimum | 0.0001 | 0.0000 | 0.0000 | 0.0000 |
| Cellobiose | Maximum | 0.2935 | 5.1987 | 3.1664 | 4.5788 |
|  | Minimum | 0.0012 | 0.0000 | 0.0000 | 0.0000 |
| Cellulose | Maximum | 0.5801 | 6.6722 | 6.2504 | 4.9125 |

the reduced network accounts for a term called "salts" gathering all the metabolites that do not participate in any other reactions, but that are, nevertheless, necessary to produce biomass. The extracellular metabolites were biomass; acetate; succinate; formate; ammonium; coenzyme A (Co-A; $CO_2$; proton (cytosolic and extracellular); ATP; salts; glycogen; $PP_i$; and the three carbon sources glucose, cellobiose, and cellulose. The reduced network (Data set S4 in supplementary material B) is appropriate for the computation of EFM.

## From EFMs to macroscopic reactions

EFMs were computed for each carbon source obtaining 9,861,037; 11,863,589; and 11,540,721 EFMs for glucose, cellobiose, and cellulose, respectively. The calculation did not consider that the network could use the three carbon sources at the same time. For the sake of analysis, only the EFMs that consumed the carbon source, produced biomass, and did not consume glycogen were considered for the analysis leading to a total of 798,872; 1,198,271; and 2,131,696 EFMs for glucose, cellobiose, and cellulose, respectively.

The computed EFMs were multiplied by the stoichiometric matrix of the extracellular metabolites to derive macroscopic reactions that can efficiently bring together metabolism and dynamics through the development of DMMs (71, 72). However, the consideration of many EFMs adds considerably to the kinetic parameters associated with substrate uptake rates leading to an overparameterization. Yield analysis (71) was presented as an alternative to perform a substantial reduction of the number of EFM from an inspection of the convex hull in a two-dimensional (2-D) representation on the yield vector space for extracellular products.

Yield analysis for the EFMs is reported for the four principal products: biomass, acetate, formate, and succinate. Yields of the computed EFMs were obtained by dividing their fluxes by the flux of the carbon source. The minima and maxima yield values obtained from EFM for the main products are reported in Table 2.

Fig. 5 displays in the diagonal the distribution of 798,872 EFMs obtained for glucose where it is observed that biomass is mainly produced at values around 0.01 g per mmol of glucose. On the other hand, most of the EFMs producing acetate, succinate, and formate report small extracellular production. The plots in the non-diagonal show the yields of the products with respect to all the products, where each blue point is an EFM. It is worth noting that the surfaces in yields mainly correspond to triangles except for the yields for formate. Similar results were obtained when the only carbon source was cellobiose (Fig. 6) and cellulose (Fig. 7).

We performed a yield analysis with a 2-D representation of the convex hull— surrounding total EFMs—to reduce the number of EFM used for macroscopic reactions of DMM. In this case, triangles were used to find a minimum number of EFM to be used in DMM. Those EFMs are denoted as red points in Fig. 8A through C which display the EFM and their reduction by yield analysis for glucose, cellobiose, and cellulose, respectively. The nine selected EFMs obtained for each substrate were compared to avoid repeated EFMs. For glucose, nine EFMs remained, while only seven and eight remained for cellobiose and cellulose, respectively (Table 3). From Table 3, macroscopic reactions for each substrate can be derived as:

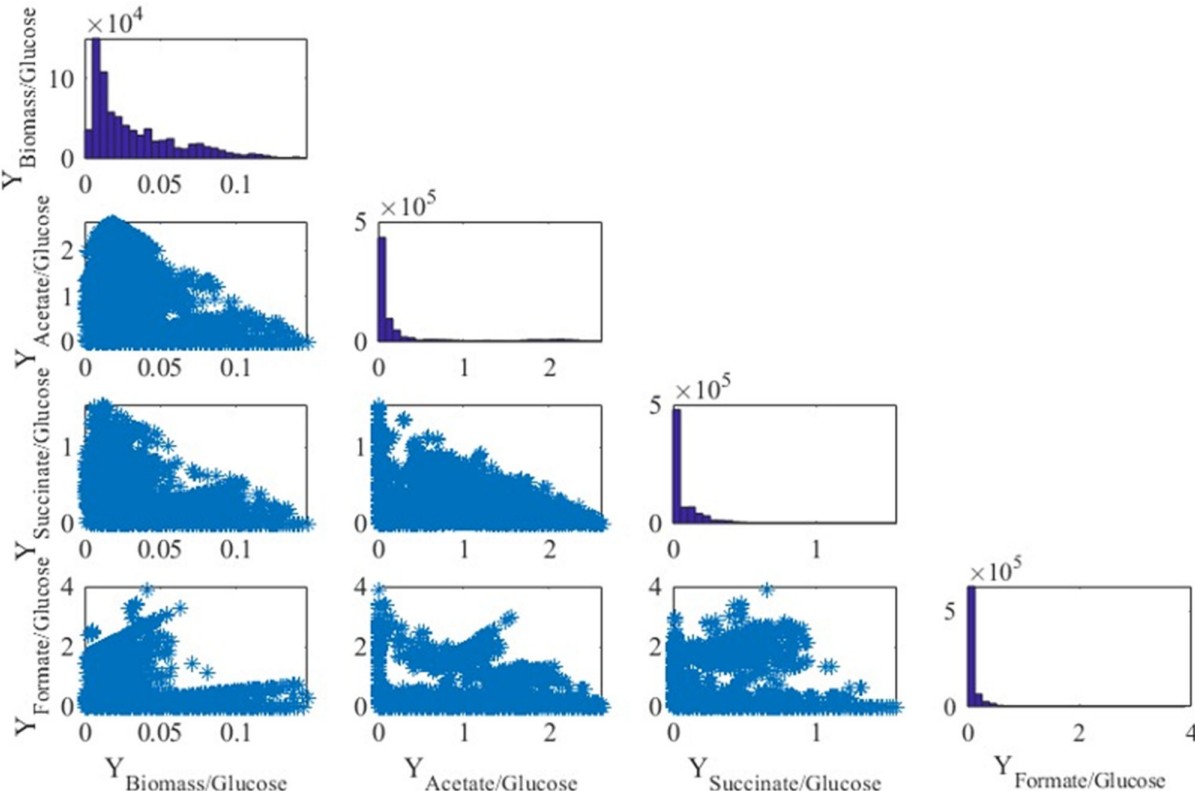

**FIG 5** Yield representation of EFM and distribution of EFM when growing on glucose. Units are millimoles per mole except for the yield biomass/glucose (gram per millimole).

$$\text{carbon substrate} + a\,\text{salts} + b\,\text{ATP} + c\,CO_2 + d\,\text{Co} - A + e\,\text{ammonium} = f\,\text{biomass} + g\,\text{acetate}$$
$$+ h\,\text{succinate} + i\,\text{formate} + j\,\text{proton}_{[c]} + k\,\text{proton}_{[e]} + l\,\text{glycogens} + m\,PP_i + n\,CO_2$$

where the coefficients $a - n$ correspond to the absolute values of the EFM, which represent the letters on the table. Note that the coefficients $c$ or $n$ will depend on whether $CO_2$ has a negative sign ($c$) or a positive sign ($n$). These EFMs are used to select a minimal set of macroscopic reactions for the DMM as discussed below.

## Dynamic metabolic model

Table 4 shows the selected EFMs and the model parameter estimates for each substrate, including the conversion factors between the biomass concentration (moles per liter) and OD for glucose and cellobiose. The metabolism of glucose and cellulose is represented by four macroscopic reactions. For cellobiose, the metabolism is represented by five macroscopic reactions. All the EFMs are from the polygon vertices of the yield spaces. The models were implemented in MATLAB and are available at https://doi.org/10.5281/zenodo.7228115.

Fig. 9A through C show the comparison of the variables predicted by the model against the dynamic experimental data obtained from the culture of *F. succinogenes* on glucose, cellobiose, and cellulose, respectively. Table 5 shows the accuracy of the model. For the experiments with glucose, the CVRMSE was 17%. For the experiments with cellobiose, the average CVRMSE was 19%. For the experiments with cellulose, the average CVRMSE was 22%.

## DISCUSSION

The mathematical modeling of the rumen ecosystem is a useful endeavor to provide tools for improving rumen function. Current kinetic rumen models do not consider

**TABLE 3** EFMs from the polygon (triangle) enclosing the yield spaces

| Glucose | Triangles | | | | | | | | | Coefficient |
|---|---|---|---|---|---|---|---|---|---|---|
| | e1 | e2 | e3 | e4 | e5 | e6 | e7 | e8 | e9 | |
| Glucose | −1 | −1 | −1 | −1 | −1 | −1 | −1 | −1 | −1 | |
| Salts | 0 | −0.006 | −0.039 | 0 | −0.004 | −0.04 | −0.043 | 0 | −0.012 | a |
| ATP | 0 | −0.008 | −0.057 | 0 | −0.005 | −0.06 | −0.064 | 0 | −0.018 | b |
| $CO_2$ | −0.361 | 0.048 | 0.333 | −0.542 | −0.751 | 0.255 | 0.372 | −0.361 | −2.166 | c/n |
| Co-A | −7.26e-7 | −8.27e-5 | −5.70e-4 | −2.26e-6 | −5.20e-5 | −5.96e-4 | −6.37e-4 | −6.91e-7 | −1.81e-4 | d |
| Ammonium | −0.001 | −0.139 | −0.956 | −0.004 | −0.087 | −0.999 | −1.068 | −0.001 | −0.304 | e |
| Biomass | 0 | 0.019 | 0.129 | 0.001 | 0.012 | 0.134 | 0.144 | 0 | 0.041 | f |
| Acetate | 0 | 2.623 | 0.414 | 1.089 | 0 | 0 | 0 | 0 | 0 | g |
| Succinate | 0.723 | 0 | 0 | 0 | 1.562 | 0.187 | 0 | 0.723 | 0.663 | h |
| Formate | 0 | 0 | 0 | 1.087 | 0 | 0 | 0 | 0 | 3.881 | i |
| Proton $_{[c]}$ | 0 | 0 | 1.138 | 0 | 2.467 | 0 | 0 | 0 | 4.806 | j |
| Proton $_{[e]}$ | 1.327 | 2.754 | 0 | 2.09 | 0 | 0 | 1.003 | 1.326 | 0 | k |
| Glycogens | 0.096 | 0 | 0.002 | 0.09 | 0.002 | 0 | 0.008 | 0.096 | 0.003 | l |
| $PP_i$ | 1.27e-5 | 1.45e-3 | 1.00e-2 | 3.96e-5 | 9.13e-4 | 1.05e-2 | 1.12e-2 | 1.21e-5 | 3.19e-3 | m |

| Cellobiose | Triangles | | | | | | | |
|---|---|---|---|---|---|---|---|---|
| | e1 | e2 | e3 | e4 | e5 | e6 | e7 | |
| Cellobiose | −1 | −1 | −1 | −1 | −1 | −1 | −1 | |
| Salts | −0.088 | 0 | −0.008 | 0 | −0.007 | 0 | −0.026 | a |
| ATP | −0.131 | 0 | −0.012 | 0 | −0.011 | 0 | −0.039 | b |
| $CO_2$ | 0.761 | −0.684 | 0 | 0 | −1.513 | −0.11 | −2.065 | c/n |
| Co-A | −1.32e-1 | −2.53e-1 | −1.18e-1 | −2.83e-6 | −1.78e-1 | −6.47e-7 | −3.90e-1 | d |
| Ammonium | −2.181 | −0.004 | −0.198 | −0.005 | −0.181 | −0.001 | −0.654 | e |
| Biomass | 0.293 | 0.001 | 0.027 | 0.001 | 0.024 | 0 | 0.088 | f |
| Acetate | 0 | 0 | 5.199 | 2.906 | 0 | 2.39 | 2.576 | g |
| Succinate | 0 | 1.37 | 0.138 | 0.003 | 3.152 | 0.148 | 0.008 | h |
| Formate | 0 | 0 | 0 | 0 | 0 | 0.074 | 4.579 | i |
| Proton $_{[c]}$ | 2.055 | 0 | 0 | 0 | 6.474 | 0 | 0 | j |
| Proton $_{[e]}$ | 0 | 2.545 | 5.327 | 2.747 | 0 | 2.575 | 6.587 | k |
| Glycogens | 0.01 | 0.2 | 0.002 | 0.171 | 0 | 0.185 | 0.029 | l |
| $PP_i$ | 2.28e-2 | 4.45e-5 | 2.07e-3 | 4.97e-5 | 1.89e-3 | 1.12e-5 | 6.84e-3 | m |

| Cellulose | Triangles | | | | | | | |
|---|---|---|---|---|---|---|---|---|
| | e1 | e2 | e3 | e4 | e5 | e6 | e7 | e8 |
| Cellulose | −1 | −1 | −1 | −1 | −1 | −1 | −1 | −1 |
| Salts | −0.049 | −0.175 | −0.001 | −0.022 | −0.029 | −0.029 | −0.028 | −0.018 | a |
| ATP | −0.072 | −0.259 | −0.001 | −0.033 | −0.043 | −0.043 | −0.041 | −0.027 | b |
| $CO_2$ | −0.204 | 1.503 | −0.391 | 0.075 | −2.602 | −0.673 | −2.215 | −3.956 | c/n |
| Co-A | −7.16e-4 | −2.57e-3 | −1.18e-5 | −3.26e-4 | −4.24e-4 | −4.31e-4 | −4.05e-4 | −2.71e-4 | d |
| Ammonium | −1.2 | −4.311 | −0.02 | −0.547 | −0.711 | −0.723 | −0.679 | −0.455 | e |
| Biomass | 0.162 | 0.58 | 0.003 | 0.074 | 0.096 | 0.097 | 0.091 | 0.061 | f |
| Acetate | 6.609 | 0 | 0 | 0 | 0 | 6.672 | 0 | 0 | g |
| Succinate | 1.244 | 0 | 0.796 | 0 | 5.699 | 1.851 | 4.904 | 4.325 | h |
| Formate | 0 | 0 | 0 | 0.231 | 0 | 0 | 0 | 3.904 | i |
| Proton $_{[c]}$ | 9.418 | 0 | 0 | 0.166 | 0 | 9.171 | 10.361 | 10.166 | j |
| Proton $_{[e]}$ | 0 | 0 | 0 | 0 | 12.064 | 0 | 0 | 0 | k |
| Glycogens | 0.001 | 0.028 | 0.586 | 0.582 | 0.007 | 0.009 | 0.089 | 0.125 | l |
| $PP_i$ | 1.26e-2 | 4.51e-2 | 2.07e-4 | 5.72e-3 | 7.44e-3 | 7.56e-3 | 7.10e-3 | 4.76e-3 | m |

genomic information (31, 65, 73, 74). GEMs are a promising tool to fill this lacking gap and allow a better understanding of the rumen systemic functionality (75) and the individual bacteria metabolism (76).

**TABLE 4** Selected EFMs of the dynamic model and parameter estimates[a]

| Glucose | | | |
|---|---|---|---|
| **e2** | **e5** | **e6** | **e9** |
| $\mu_{\text{max,i}}(\text{h}^{-1})$ 0.037 | 0.031 | 0.31 | 0.004 |
| $K$ (M) $9 \times 10^{-3}$ | | | |
| $\alpha(\text{M/OD})$ $6.15*10^{-4}$ | | | |

| Cellobiose | | | | |
|---|---|---|---|---|
| **e1** | **e4** | **e5** | **e6** | **e7** |
| $\mu_{\text{max,i}}(\text{h}^{-1})$ 0.33 | 0.0002 | 0.033 | 0.0001 | 0.007 |
| $K$ (M) $9*10^{-3}$ | | | | |
| $\alpha(\text{M/OD})$ $7.24*10^{-4}$ | | | | |

| Cellulose | | | |
|---|---|---|---|
| **e1** | **e3** | **e6** | **e8** |
| $\mu_{\text{max,i}}(\text{h}^{-1})$ $0.087*10^{-3}$ | $0.34*10^{-3}$ | $0.96*10^{-3}$ | $0.08*10^{-3}$ |
| $K_c(\text{mol}_{\text{cellulose}}/\text{mol}_{\text{biomass}})$ 6.05 | | | |

[a]The stoichiometry of the EFMs is given in Table 3.

**TABLE 5** Model accuracy

| Glucose utilization | | | | | |
|---|---|---|---|---|---|
| | Acetate | Succinate | Formate | Ammonia | Substrate | OD |
| 100*CVRMSE[a] | 34 | 19 | 11 | 10 | 14 | 14 |

| Cellobiose utilization | | | | | |
|---|---|---|---|---|---|
| | Acetate | Succinate | Formate | Ammonia | Substrate | OD |
| 100*CVRMSE[a] | 18 | 11 | 19 | 8 | 44 | 17 |

| Cellulose utilization | | |
|---|---|---|
| | Acetate | Succinate | Formate |
| 100*CVRMSE[a] | 15 | 38 | 13 |

[a]Coefficient of variation of the RMSE (CV(RMSE)).

Many independent methods have been developed to generate genome-scale models, including some toolboxes and workspaces, such as Pathway Tools (39), RAVEN (77), merlin (78), KBase (79), The SEED (80), AuReMe (38), AutoKEGGRec (67), CarVeMe (81), and gapseq (82). They rely on one or several metabolic databases such as MetaCyc (48), KEGG (83), ModelSEED (84), or BiGG (68). However, the output of a main platform for a GEM requires adjustments assisted by a choice of specialized tools, especially when the network reconstruction requires to take advantage of information spread in different models, formats, and organisms, leading to issues in the standardization of metadata and reproducibility of the reconstruction procedure. In this work, the GEM construction of *F. succinogenes* S85 was performed using the AuReMe platform, selected for its full traceability reconstruction (38) and capabilities to produce high-quality reconstructions (85).

GEMs are widely used for microbial-defined growth medium identification (30, 86), metabolic functional characterization (29, 87), or the design of novel treatment against pathogens (88). Regarding gut communities, they have been mainly applied to human gut bacteria to decipher the microbial interactions in the human intestinal microbiome (28, 30, 89–91).

Until now there have been few GEMs available for rumen bacteria such as the networks for the lactate-utilizing bacterium *Megasphaera elsdenii* (76), and the succinic acid–producing strain *Actinobacillus succinogenes* (92). Recently, one simplified representative rumen community metabolic model was reported (75). A synthetic community composed of a cellulolytic bacterium, a proteolytic bacterium, and a methanogen was developed to enlighten metabolite secretion profiles, community compositions, and

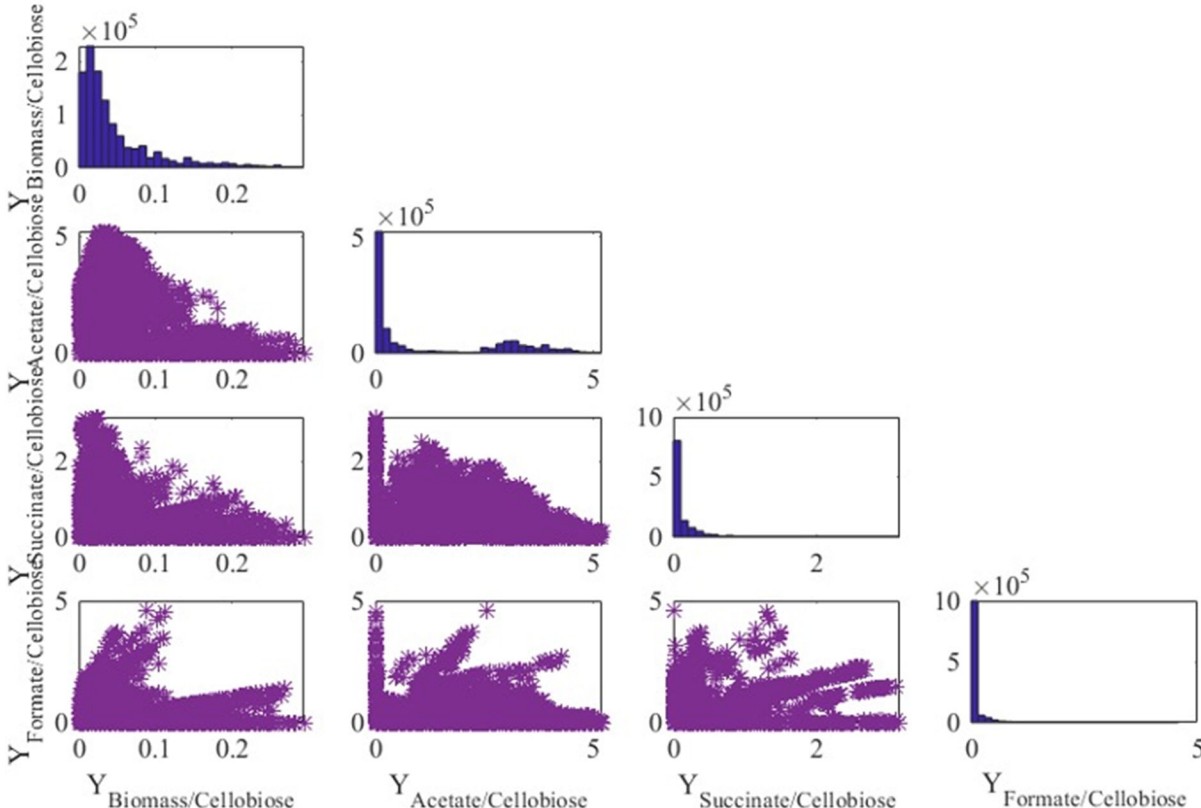

**FIG 6** Yield representation and distribution of EFM when growing on cellobiose. Units are millimoles per mole except for the yield biomass/cellobiose (gram per millimole).

interactions with bacteriophages (75). Our work contributes to expand the application of the GEM approach to study the rumen ecosystem.

The obtained GEM of *F. succinogenes* is composed of thousands of metabolites and reactions associated with their genes and can be set as a useful network information for generating future ruminal bacterial draft models. The 1,317 genes of the final *F. succinogenes* S85 model cover more than 41.5% of the genes identified in the two genomes of the strain (3,170 and 3,161 ASM14650v1 and ASM2466v1, respectively).

Our model contains 2.5 times more genes than the model of the rumen cellulolytic bacterium *Ruminococcus flavefaciens* previously reconstructed (75) by ModelSEED (84) and gaps-filled by GapFind-GapFill (75), as well as 1.5 more reactions and metabolites. Our final reconstructed network is functional for biomass and SCFA production with compacted core metabolism represented by 30% of active pathways (Data set S3 in supplementary material B) and has a simulated growth rate of 72% times greater than that of *R. flavefaciens*, its cellulolytic model candidate (75). GEMs are very powerful to provide a qualitative analysis of microbial metabolism. However, they are limited to quantitative prediction of the dynamics of metabolites. This work develops an approach for developing a DMM exploiting the microbial genomic information embedded in the GEM of *F. succinogenes* S85. The use of GEM for dynamic modeling and other analysis methods is cumbersome due to the large number of reactions and metabolites, which hamper the interpretation/visualization of fluxes (e.g., FBA) and limit the calculation of computationally expensive analysis (e.g., EFM) (93). Hence, a reduction of the GEM into a network that still captures phenotypic and genotypic properties while displaying flexibility is needed (94). For our modeling exercise, the NetRed tool (58) was instrumental to perform the reduction of our network. All the full-scale GEM reactions participating in the 63 reduced-scale genome-based metabolic network reactions of *F. succinogenes*

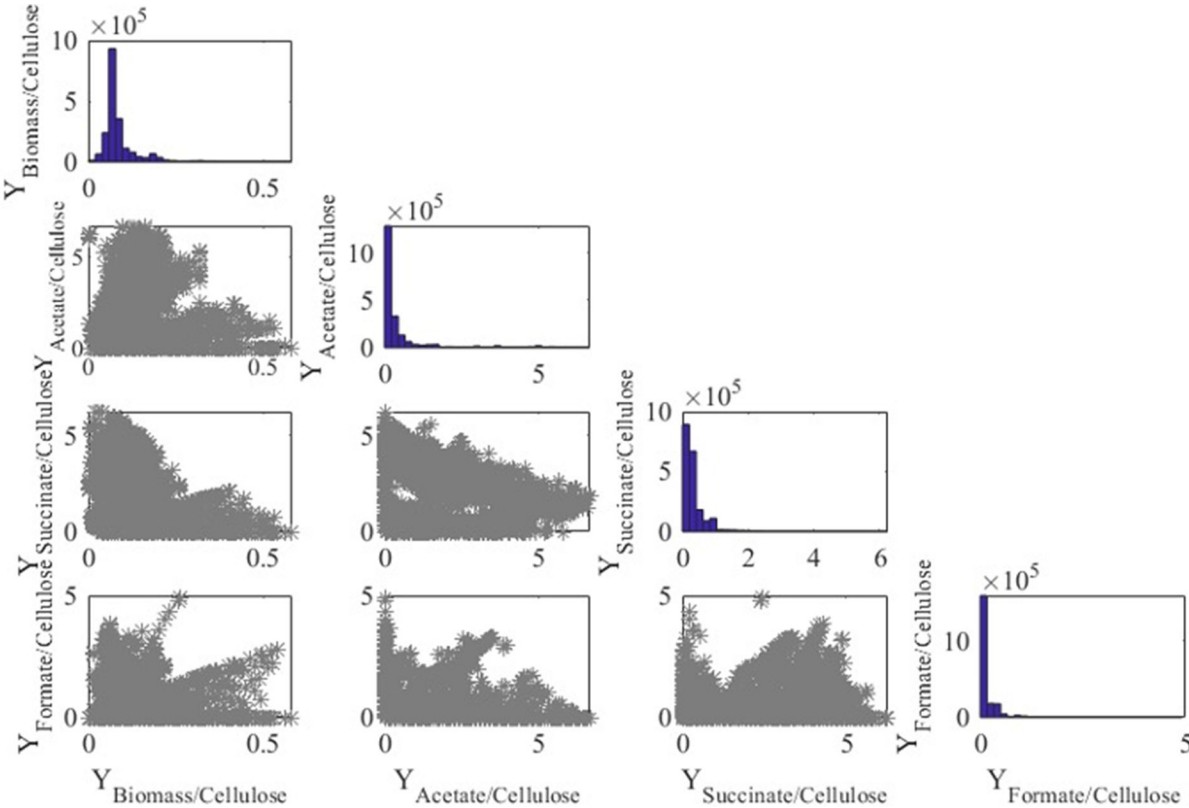

**FIG 7** Yield representation and distribution of EFM when growing on cellulose. Units are millimoles per mole except for the yield biomass/cellulose (gram per millimole).

S85 are present in the list of active pathways of the large-scale network (Data set S2 in supplementary material B).

Yield analysis was used in the flux vectors to verify the carbon balance of the network. The obtained yields showed that carbon balance was not respected, and the difference was coming from the glycogen pathways. Some of the glycogen pathway reactions were identified from the databases as generic reactions. *F. succinogenes* is known to synthesize glycogen during all its growth phases (19). We focused on the validation of its production/degradation pathway including generic reactions not only for its specificity in this organism but also for its importance in the carbon cycle equilibrium that needs to be held for the network, the EFM computation step, and the DMM. As a solution, we have set for glycogen a number of monomers equal to 6 to be able to complete the carbon balance and therefore stoichiometrically balance the reactions of this metabolic pathway. The reduced network helped to calculate the EFM whose reduction was performed by yield analysis (71). A 2-D representation of yields was used to compute the convex hull that surrounds the EFMs. The EFMs belonging to the convex hull were further reduced by a method employing polygons (95). The EFMs on the convex hull are normally chosen to provide a wide range of steady states to the system.

The resulting model structures are similar in degree of complexity with respect to the rumen fermentation model developed by Muñoz-Tamayo et al. (64), where carbohydrate metabolism is represented by five macroscopic reactions. The main difference in the approach developed in the present work is that the macroscopic reactions are derived from the reconstructed metabolic network of *F. succinogenes*. It should be noted that the resulting macroscopic reactions included in the models correspond to the active sets of EFMs specific to the experimental here studied while the subsets of all EFMs at the vertices of the polygon enclosing the yield spaces shown in Fig. 8A through C constitute a minimal generating of EFMs covering almost all possible metabolic states. This

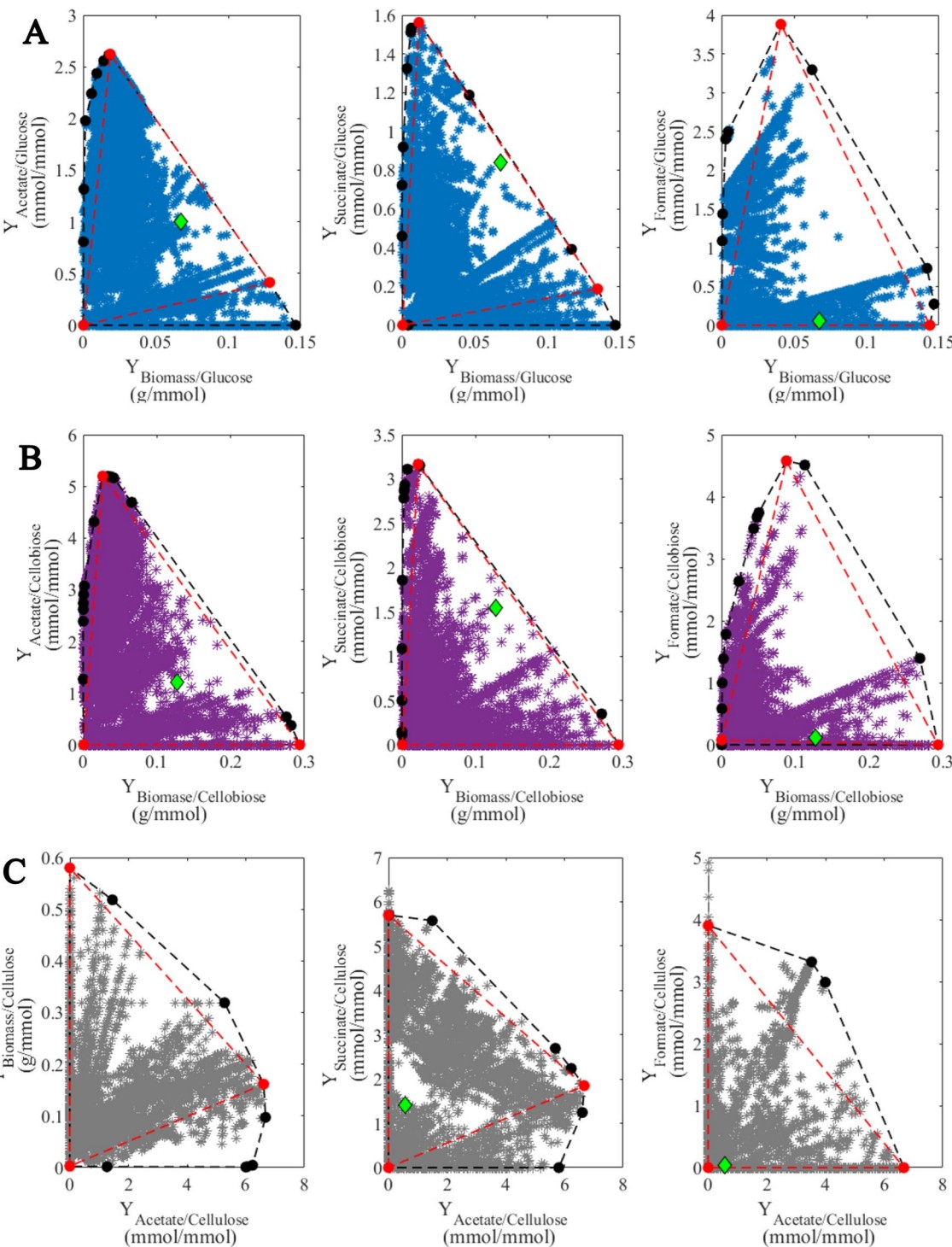

**FIG 8** Yield analysis of the EFM for A (blue stars) glucose, B (purple stars) cellobiose, and C (gray stars) cellulose. Representation computation of the convex hull (black circles) and reduction of the convex hull (red circles) with respect to experimental data (diamonds).

approach provides a high flexibility to span the metabolic space at different experimental conditions. Such a flexibility in the model structure is a great asset to study in the future strategies to enhance substrate utilization and target-desired fermentation profile.

The model performances were acceptable to capture the dynamics of fermentation by *F. succinogenes*. However, Table 5 and Fig. 9A through C display that there is room for improvement. The prediction of succinate by the model for cellulose utilization has

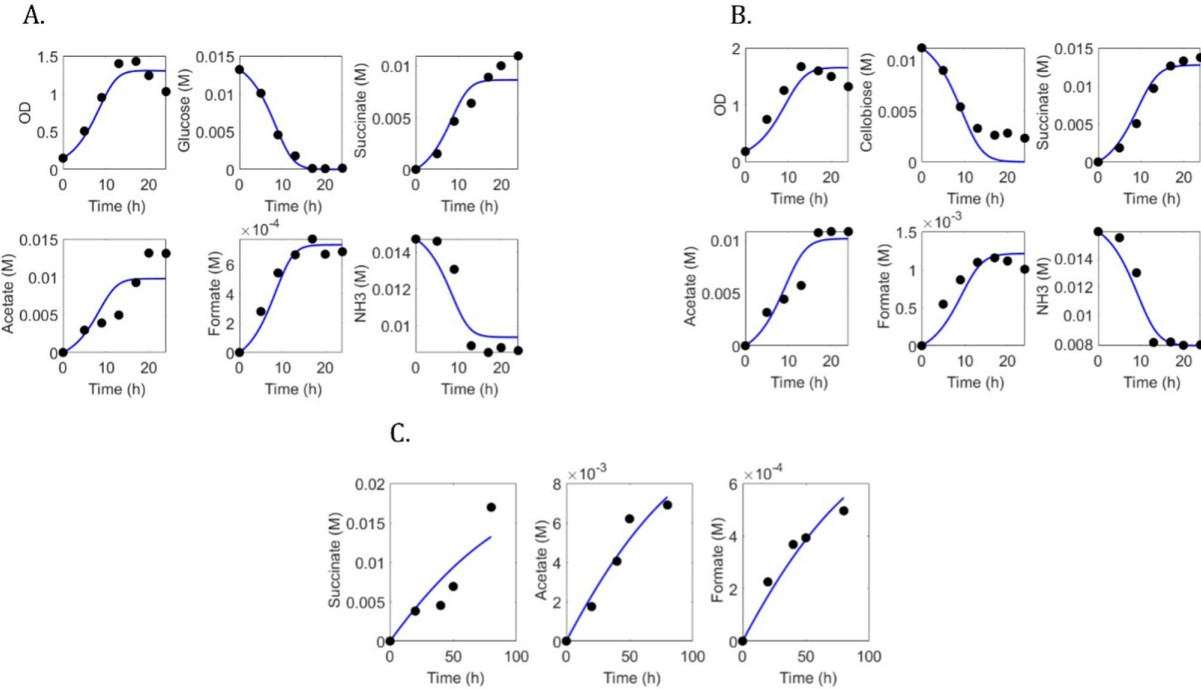

**FIG 9** Experimental data (•) of the fermentation of (A) glucose, (B) cellobiose, and () cellulose by *Fibrobacter succinogenes* S85 compared against the variables predicted by the dynamic models (solid line).

indeed a high CVRMSE. One key element for model improvement is glycogen metabolism, which was not integrated into this work. Glycogen plays an important role in *F. succinogenes* and appears to be submitted to a futile cycle that results from simultaneous utilization and storage (19, 96). As observed in Table 3, glycogen is a net product for the EFMs of the polygon vertices. Thereby, the current model structure cannot account for the futile cycling. This limitation is intrinsic to the steady-state assumption for the EFM derivation. To account for glycogen futile cycling, it will be then required to split the metabolic network into subnetworks. The procedure of network splitting can be done on the knowledge basis as applied for example to study microalgae metabolism (97). However, the splitting method is a challenging issue. As a perspective, in the mid-term, we will explore the use of splitting techniques such as those developed by Verwoerd (98) and Schuster (99) to account for the glycogen futile cycle. In the long term, we will apply the approach here developed to other key rumen microbes to address the modeling of rumen microbial mini consortia. As we have previously discussed (100), this approach will enable us to construct tractable models that integrate genomic information with capabilities to inform on strategies for driving the rumen microbiome.

## ACKNOWLEDGMENTS

We thank the INRAE PHASE department and the INRAE MEM metaprogram for financial support. The Ph.D. of Ibrahim Fakih is supported by grants from the INRAE Holoflux metaprogram and funding from Lallemand Animal Nutrition (Blagnac, France).

We thank Emile Dumont and Méziane Aite for their preliminary work on network reconstruction and Melanie Brunel for the cultivation experiments with *F. succinogenes*.

The authors declare that they have no competing interests.

## AUTHOR AFFILIATIONS

[1]Université Clermont Auvergne, INRAE, UMR454 Microbiologie Environnement Digestif et Santé, 63000 Clermont-Ferrand, France

²Université Paris-Saclay, INRAE, AgroParisTech, UMR Modélisation Systémique Appliquée aux Ruminants, 91120 Palaiseau, France
³Université Rennes, Inria, CNRS, IRISA, Dyliss team, 35042 Rennes, France
⁴TBI, Université de Toulouse, CNRS, INRAE, INSA, Toulouse, France

## AUTHOR ORCIDs

Ibrahim Fakih  http://orcid.org/0000-0001-6202-062X
Jeanne Got  http://orcid.org/0000-0002-2310-0843
Carlos Eduardo Robles-Rodriguez  http://orcid.org/0000-0002-2436-3653
Anne Siegel  http://orcid.org/0000-0001-6542-1568
Evelyne Forano  http://orcid.org/0000-0002-7450-3466

## FUNDING

| Funder | Grant(s) | Author(s) |
| --- | --- | --- |
| Institut National de Recherche pour l'Agriculture, l'Alimentation et l'Environnement (INRAE) | | Ibrahim Fakih |
| Lallemand Animal Nutrition | | Ibrahim Fakih |

## AUTHOR CONTRIBUTIONS

Ibrahim Fakih, Conceptualization, Data curation, Formal analysis, Investigation, Methodology, Writing – original draft | Jeanne Got, Conceptualization, Data curation, Formal analysis, Methodology, Software, Writing – review and editing | Carlos Eduardo Robles-Rodriguez, Conceptualization, Data curation, Formal analysis, Investigation, Methodology, Software, Writing – review and editing | Anne Siegel, Conceptualization, Formal analysis, Methodology, Supervision, Writing – review and editing | Evelyne Forano, Conceptualization, Formal analysis, Funding acquisition, Methodology, Supervision, Writing – review and editing | Rafael Muñoz-Tamayo, Conceptualization, Formal analysis, Funding acquisition, Methodology, Project administration, Software, Supervision, Writing – review and editing

## DATA AVAILABILITY STATEMENT

The supplementary material of the article, the metabolic networks, and mathematical models developed in this work are available at https://doi.org/10.5281/zenodo.7228115.

## ADDITIONAL FILES

The following material is available online.

### Open Peer Review

**PEER REVIEW HISTORY (review-history.pdf).** An accounting of the reviewer comments and feedback.

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
