## [Reviewer comments · mSystems]

Dynamic genome-based metabolic modeling of the predominant cellulolytic rumen bacterium *Fibrobacter succinogenes* S85

Ibrahim Fakh, Jeanne Got, Carlos Robles-Rodriguez, Anne Siegel, Evelyne Forano, and Rafael Munoz-Tamayo/Rafael

Corresponding Author(s): Rafael Munoz-Tamayo/Rafael, Institut National de Recherche pour l'Agriculture l'Alimentation et l'Environnement

Review Timeline:

Submission Date:	October 24, 2022
Editorial Decision:	February 17, 2023
Revision Received:	March 3, 2023
Accepted:	March 14, 2023

Editor: Vanni Bucci

Reviewer(s): Disclosure of reviewer identity is with reference to reviewer comments included in decision letter(s). The following individuals involved in review of your submission have agreed to reveal their identity: Chao Ye (Reviewer #2)

Transaction Report:

DOI: <https://doi.org/10.1128/msystems.01027-22>

February 17, 2023

Dr. Rafael Munoz-Tamayo/Rafael
Institut National de Recherche pour l'Agriculture l'Alimentation et l'Environnement
Paris
France

Re: mSystems01027-22 (Dynamic genome-based metabolic modeling of the predominant cellulolytic rumen bacterium *Fibrobacter succinogenes* S85)

Dear Dr. Rafael Munoz-Tamayo/Rafael:

Thank you for submitting your manuscript to mSystems. We have completed our review and I am pleased to inform you that, in principle, we expect to accept it for publication in mSystems. However, acceptance will not be final until you have adequately addressed the reviewer comments.

Preparing Revision Guidelines

Sincerely,

Vanni Bucci

Editor, mSystems

Journals Department
American Society for Microbiology
1752 N St., NW

Reviewer comments:

Reviewer #1 (Comments for the Author):

Manuscript mSystems01027-22 reports the construction of genome-scale metabolic model for the predominant cellulolytic rumen bacterium *Fibrobacter succinogenes* S85. The model was further reduced to calculate EFMs. Fermentations of glucose, cellobiose and cellulose were simulated by dynamic metabolic modelling. This study is of a certain level of novelty as it reports the first model for this important bacterium.

Some minor issues:

Line 23: is "931 pathways" correct?

Line 142: Can *E. coli* biomass reaction be used in *F. succinogenes*? To what extent the difference of biomass composition of *F. succinogenes* would impact the modelling accuracy?

Line 153, wrong name for *Bacteroides thetaiotaomicron*, no strain name?

Line 217-221: Any mistakes in this sentence "NetRed is based on matrix algebra taking as inputs the stoichiometric matrix, a flux vector, the numerical flux values, and a list of protected metabolites, and it is implemented in the MATLAB COBRA toolbox"? Also, the expression regarding advantages of NetRed is confusing. Please clarify these two sentences, in particular, specify the advantages of using NetRed rather than other network reduction methods.

Line 226: what are those "different objective functions and several input fluxes"? Please clarify them.

Line 247: Variable B represents the biomass molar concentration according to authors explanation. How was it calculated? The unit of the left side of Eq (3) is h⁻¹, while it is mol·h⁻¹ at the right side. Therefore it is wrong.

Line 251-252: Can authors provide a bit more details about high correlation of Monod parameters? Why is that a problem and what is the potential impact? From current expression, one from fields other than this area may feel difficult to understand it.

Line 261: where is value of the conversion factor?

Lines 172, 182 and 297: should be "BiGG" database

Reviewer #2 (Comments for the Author):

(1) In the background part, there is a little introduction to the dynamic metabolism model, please introduce it in detail.

(2) There were problems with writing, some formats and units were wrong. So please revise it carefully.

a) Please improve the clarity of the figures so that the text in the figures is clearly visible.

b) The figures in the text are represented as Fig. X, but the illustration of the figure is Figure X. Please standardize the format.

c) Please align the caption in figure 1 with the others.

d) There is no need to use italics for the figure illustration.

e) Line 321 "38,5%" was wrong.

(3) The authors need to write the methods and materials section in more detail. Such as the culture medium preparation and how you measure the concentrations of succinate, acetate, formate, ammonia, and glucose at six time points in culture supernatants. What are the six time points? How many replicates were done? What is the method of the simulated growth rate?

(4) In line 175, why proteins with more than 76% homology are related to their corresponding reactions in GEM? How did 76% come from?

(5) In line 310, the description of "68 of those were added by Gap-filling" does not match Figure 2A.

(6) In figure 9, the authors show the comparison of the experimental data against the variables predicted by the model. But where they got the in vitro data should be clear. Can the author describe it in detail instead of just one sentence?

Dear Dr. Vanni Bucci and reviewers,

Thank you for the evaluation of our manuscript. We addressed all comments and suggestions in the revised article. Below are the responses to the comments. We hope you find the paper suitable for publication in mSystems.

Rafael Muñoz-Tamayo

Reviewer 1

Manuscript mSystems01027-22 reports the construction of genome-scale metabolic model for the predominant cellulolytic rumen bacterium *Fibrobacter succinogenes* S85. The model was further reduced to calculate EFMs. Fermentations of glucose, cellobiose and cellulose were simulated by dynamic metabolic modelling. This study is of a certain level of novelty as it reports the first model for this important bacterium.

R. Thank you for your feedback.

Line 23: is "931 pathways" correct?

R. We double checked the list of pathways and it is 931 pathways. The list is available on the supplementary data.

Line 142: Can *E. coli* biomass reaction be used in *F. succinogenes*? To what extent the difference of biomass composition of *F. succinogenes* would impact the modelling accuracy?

R. The reviewer is right on pointing out this aspect. It should be noted that we used the biomass reaction of *E. coli* as template since biomass formation involves components that are common for Bacteria. However, the reaction was modified to account for the specificities of *F. succinogenes* metabolism. We added a clarification in L149.

Line 153, wrong name for *Bacteroides thetaiotaomicron*, no strain name?

R. The strain name was added: L160.

Line 217-221: Any mistakes in this sentence "NetRed is based on matrix algebra taking as inputs the stoichiometric matrix, a flux vector, the numerical flux values, and a list of protected metabolites, and it is implemented in the MATLAB COBRA toolbox"? Also, the expression regarding advantages of NetRed is confusing. Please clarify these two sentences, in particular, specify the advantages of using NetRed rather than other network reduction methods.

R. The sentence was reformulated and additional information about the advantages of NetRed has been added in L223-232.

Line 226: what are those "different objective functions and several input fluxes"? Please clarify them.

R. Information was added: L236-240.

Line 247: Variable B represents the biomass molar concentration according to authors explanation. How was it calculated? The unit of the left side of Eq (3) is h^{-1} , while it is $mol \cdot h^{-1}$ at the right side. Therefore it is wrong.

R. For construction, the unit of biomass is given in moles/L since the stoichiometry matrix is set by molar units. However, the reviewer is right in questioning how these units are applied in practice. In our experiments, biomass was measured in optical density units. To integrate the measurements in our model, we had to estimate a conversion factor between mol/L and OD. The information is clarified in L283-286. Note that the conversion factor is displayed in the Matlab scripts of the model.

Line 251-252: Can authors provide a bit more details about high correlation of Monod parameters? Why is that a problem and what is the potential impact? From current expression, one from fields other than this area may feel difficult to understand it.

R. Complementary information was added: L267-274.

Line 261: where is value of the conversion factor?

R. The values were reported in the Matlab scripts. For clarity, we added the values in Table 4 and a sentence in L466-467.

Lines 172, 182 and 297: should be "BiGG" database

R. Changes were done

Reviewer #2

R. Thank you for your comments.

(1) In the background part, there is a little introduction to the dynamic metabolism model, please introduce it in detail.

R. Complementary information was added together with dedicated literature: L103-107.

(2) There were problems with writing, some formats and units were wrong. So please revise it carefully.

R. We have checked and corrected all the pointed problems.

a) Please improve the clarity of the figures so that the text in the figures is clearly visible.

R. Figures were improved with higher resolution

b) The figures in the text are represented as Fig. X, but the illustration of the figure is Figure X. Please standardize the format.

R. Done.

c) Please align the caption in figure 1 with the others.

R. Done.

d) There is no need to use italics for the figure illustration.

R. We keep the italics when relevant

e) Line 321 "38,5%" was wrong.

R. Correction was done: L343.

(3) The authors need to write the methods and materials section in more detail. Such as the culture medium preparation and how you measure the concentrations of succinate, acetate, formate, ammonia, and glucose at six time points in culture supernatants. What are the six time points? How many replicates were done? What is the method of the simulated growth rate?

R. As suggested by the reviewer we clarified the substrate and end-products assays methodology, as well as we introduced the media composition (L113-117) and the six sampling times of the bacterial growth (L121-123). The number of replicates is now also shown in L125-126. The methods for metabolite determination are in the section 2.2 (L125-130).

(4) In line 175, why proteins with more than 76% homology are related to their corresponding reactions in GEM? How did 76% come from?

R. The percent homology is often used to mean sequence identity. Different studies establish >40% identical enzyme sequences as homologs (Addou et al. 2009). We decided to be more stringent as other studies (Gontang et al. 2010) since false link of a reaction to a gene can have an impact in the model accuracy or for further biological usage. We limited gene linkage only to high identity percentage with coverage throughout the sequence. We added a sentence in L180-182.

(5) In line 310, the description of "68 of those were added by Gap-filling" does not match Figure 2A.

R. Thank you for pointing this out. The 68 reactions are part of the reactions identified by gap filling and are not linked to genes. The sentence was reformulated: L332.

(6) In figure 9, the authors show the comparison of the experimental data against the variables predicted by the model. But where they got the in vitro data should be clear. Can the author describe it in detail instead of just one sentence?

R. Information was added in L253-258, L471-473.

References

- Addou S, Rentzsch R, Lee D, Orengo CA. 2009. Domain-Based and Family-Specific Sequence Identity Thresholds Increase the Levels of Reliable Protein Function Transfer. *Journal of Molecular Biology* 387:416–430.
- Gontang EA, Gaudêncio SP, Fenical W, Jensen PR. 2010. Sequence-Based Analysis of Secondary-Metabolite Biosynthesis in Marine *Actinobacteria*. *Appl Environ Microbiol* 76:2487–2499.

March 14, 2023

Dr. Rafael Munoz-Tamayo/Rafael
Institut National de Recherche pour l'Agriculture l'Alimentation et l'Environnement
Paris
France

Re: mSystems01027-22R1 (Dynamic genome-based metabolic modeling of the predominant cellulolytic rumen bacterium *Fibrobacter succinogenes* S85)

Dear Dr. Rafael Munoz-Tamayo/Rafael:

Your manuscript has been accepted, and I am forwarding it to the ASM Journals Department for publication. For your reference, ASM Journals' address is given below. Before it can be scheduled for publication, your manuscript will be checked by the mSystems production staff to make sure that all elements meet the technical requirements for publication. They will contact you if anything needs to be revised before copyediting and production can begin. Otherwise, you will be notified when your proofs are ready to be viewed.

If you would like to submit a potential Featured Image, please email a file and a short legend to msystems@asmusa.org. Please note that we can only consider images that (i) the authors created or own and (ii) have not been previously published. By submitting, you agree that the image can be used under the same terms as the published article. File requirements: square dimensions (4" x 4"), 300 dpi resolution, RGB colorspace, TIF file format.

We recognize that the video files can become quite large, and so to avoid quality loss ASM suggests sending the video file via <https://www.wetransfer.com/>. When you have a final version of the video and the still ready to share, please send it to mSystems staff at msystems@asmusa.org.

Sincerely,

Vanni Bucci
Editor, mSystems

Journals Department
E-mail: mSystems@asmusa.org